# Pseudomonas syringae pv. tomato and the fall armyworm modulate the morpho-physiology and the metabolome of potato plants

Sandra Maluleke[1], Udoka Vitus Ogugua[1], Ntakadzeni Edwin Madala[2], Phumzile Sibisi[1], Mbalenhle Precious Mazibuko[1], Ebere Lovelyn Udeh[1], Robert Sicelo Nofemela[3], Khayalethu Ntushelo[1]*

1 Department of Agriculture and Animal Health, University of South Africa, Florida, South Africa,
2 Department of Biochemistry and Microbiology, University of Venda, Thohoyandou, South Africa,
3 Agricultural Research Council, Plant Health and Protection, Queenswood, South Africa

* ntushk@unisa.ac.za

## Abstract

Potato seedlings were challenged with two parasites, namely, the fall armyworm and *Pseudomonas syringae* pv. *tomato*, in combination and individually. Growth (plant height, stem diameter, total number of tubers and total tuber weight) and physiological function (photosynthesis rate, stomatal conductance, transpiration efficiency, the ratio of intercellular $CO_2$ concentration to ambient $CO_2$ concentration (Ci/Ca) and water use efficiency) were measured to assess the effects of the parasites on the plants. A correlation analysis of the measured growth and physiology parameters was done to understand the co-ordination of the parasite-attacked plant processes. Finally, plant metabolomic profiles were determined to assess the effects of the parasites on the metabolomes of the treated plants. Individually and in combination, the parasites had varied effects on the growth and physiology of the plants. The correlation analysis also revealed key associations between the growth and physiology aspects, and the parasites caused metabolomic reprogramming in the treated plants. Some of the results were expected but there were also unexpected outcomes. Surprisingly, the pest drastically reduced plant height when administered alone, but its ability to reduce height lessened when it was co-administered with the bacterium. The lessened ability of the pest to reduce plant height in the presence of the bacterium hints at parasite-to-parasite antagonism. This same pattern extended to stem diameter and total tuber weight. The pest individually reduced stem diameter and total tuber weight, but not when co-administered. This also hints at parasite-to-parasite antagonism. However, this matter warrants further investigation. In conclusion, the pest and the pathogenic bacterium induce morpho-physiological and metabolomic changes in potato seedlings, their effects on the measured parameters vary, and there is a possible parasite-to-parasite antagonism.

**Data availability statement:** All relevant data are within the paper and its Supporting Information files.

**Funding:** This study was partially funded by the National Research Foundation of South Africa under the grant TTK170413227119.

**Competing interests:** The authors have declared that no competing interests exist.

## Introduction

Potato (*Solanum tuberosum* L.) is a widely grown crop after the major grain crops rice, wheat, and maize [1]. Potato is grown for its starchy tubers which are also rich in fibre, protein and vitamins. The tuber also possesses key nutrients like polyphenols, carotenoids, and tocopherols. The potato crop is ecologically adaptable in most major crop producing areas of the world. In 2021, worldwide production of potatoes exceeded 370 million tonnes with China and India producing over a third of the total produce [2]. On the downside, the potato crop is prone to various pests and diseases which reduce yield and degrade tuber quality. Among some of the major threats to the potato plant is the fall armyworm (FAW), *Spodoptera frugiperda* (J.E. Smith) (Lepidoptera: Noctuidae). This lepidopteran pest punches leaves of host plants and, if severe, leaves them skeletonized with just the midrib and major veins. Severe damage collapses the foliage and eventually the plant dies. FAW is native to the Americas from where it has spread widely throughout the world and appearing in Africa for the first time in 2016 [3]. The FAW consists of two genetically differential strains known as the corn strain (C-strain) and rice strain (R-strain). The C-strain is found more commonly on corn, cotton, and sorghum, whereas the R-strain is found on rice and grasses [4]. This pest has a wide host range of more than 350 plant species [5] and in South Africa it caused substantial damage to the maize crop. According to Makgoba et al. [6], the FAW caused most damage on the leaves of the maize plant, followed by heads and ears. Its effect on the potato crop is less pronounced. The first record of the FAW damage on potatoes was in Pakistan in 2021 [7]. Feeding caused a visible single hole at the base of the stem resulting in a sudden decline of the feeding tiller [7]. Another threat to the potato plant is foliar bacterial pathogens. An important model for studying plant-bacterial interactions, the foliar bacterial pathogen, *Pseudomonas syringae* pv. *tomato*, infects the plant by producing effectors with its type III secretion system. *P. syringae* pv. *tomato* is not commonly found in potatoes but in tomatoes and causes bacterial speck [8], a devastating disease characterized by small, irregular and dark brown necrotic spots usually with a yellow halo [9]. *P. syringae* pv. *tomato* causes yield and crop loss under severe conditions [10]. Disease outbreak is favoured by high leaf moisture, cool temperatures and cultural practices that allow the bacteria to be disseminated between host plants [11]. In 1991, a *P. syringae* pv. *tomato* strain, DC3000, was reported to infect not only its natural host tomato but also Arabidopsis in the laboratory [12]. The genomic analysis of *P. syringae* pv. *tomato* strain DC3000 indicates that this strain carries a large range of potential virulence factors which include proteinaceous effectors that are secreted through the type III secretion system and a polyketide phytotoxin called coronatine [13]. Several studies uncovered the effect of *P. syringae* pv. *tomato* in tomato plants' growth and development but the effect of this bacterium on potato growth, physiology and the metabolome remains unknown. Although *P. syringae* pv. *tomato* is not a significant potato pathogen, the plethora of information on already accumulated on *P. syringae* pv. *tomato* can augment our understanding of potato plant responses to parasitism. It is also important not to study one aspect of response to parasitism but

the entire package of growth, physiology and metabolomics. The purpose of this study was therefore to assess the effect of the fall armyworm and *P. syringae* pv. *tomato* on the growth, physiology and the metabolome of potato plants. Because of the taxonomic closeness of potato to tomato, and the fact that *P. syringae* pv. *tomato* attacks the distant Arabidopsis, the authors of the current manuscript reasoned that potato seedlings should be prone to *P. syringae* pv. *tomato* especially if it is introduced by pressure infiltration. As a model bacterium, it was also important to select *P. syringae* pv. *tomato* to test on potato plants to understand potato-bacterial interactions.

## Materials and methods

Potato seedlings (cv. Georgina, G2) were challenged with the FAW and the foliar bacterial pathogen *P. syringae* pv. *tomato*. Growth and photosynthesis measurements were taken to assess the effect of the two parasites. The obtained measurements were further correlated to assess dependency of plant processes. Furthermore, metabolite profiling of the sampled leaves was done. Work was done following the graphical abstract (created using www.BioRender.com) and time schedule shown on Figs 1 and 2.

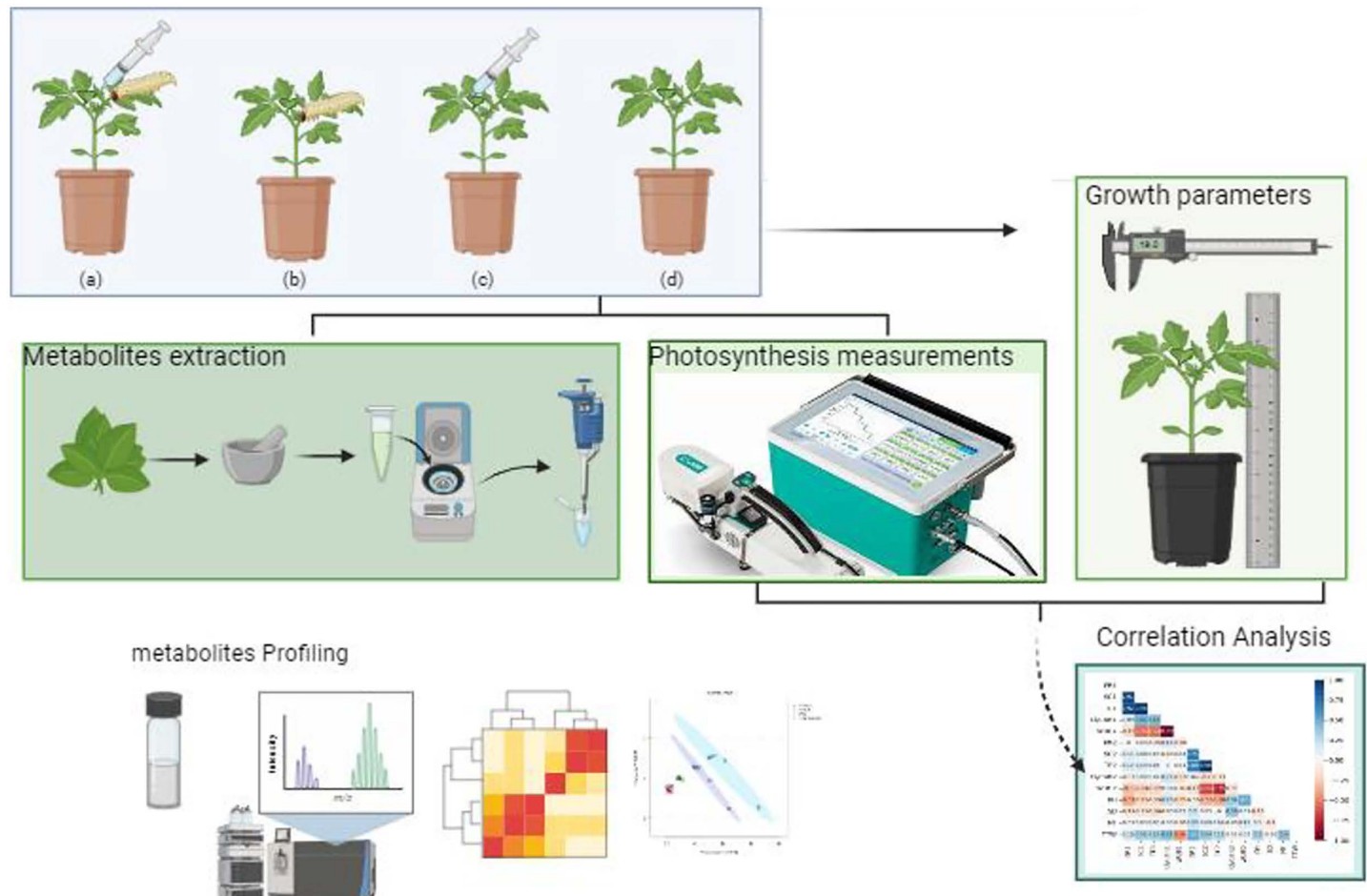

**Fig 1. Graphical abstract illustrating the workflow from plant infestation with the fall armyworm and *Pseudomonas syringae* pv. *syringae*.** Growth, photosynthesis measurements and metabolite profiling were done. Finally, a correlation analysis of the growth and photosynthesis measurements was conducted.

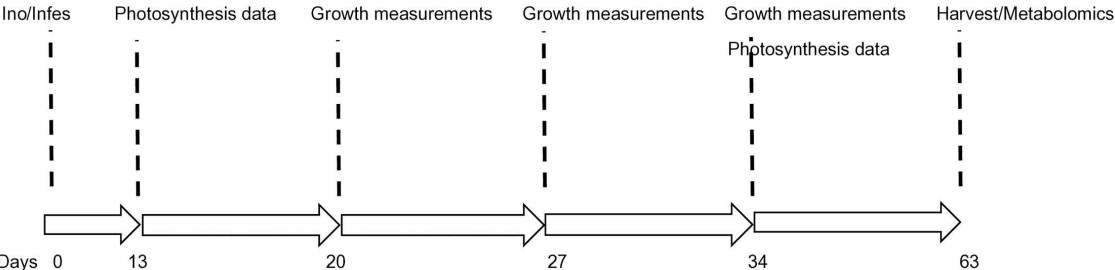

**Fig 2. Timeline of the experiment.** At day 0 the plants were inoculated with *Pseudomonas syringae* pv. *tomato* (strain BD2110) and infested with *Spodoptera frugiperda*. At day 13 and day 34 photosynthesis measurements were taken, and growth was measured at days 20, 27, and 34. Finally, the plants were harvested at day 63 and for each plant the tubers were weighed and counted. Secondary metabolome profiling was also done at day 63.

## Plant material

Twenty-one potato seedlings (cv. Georgina, G2) maintained in a greenhouse at 25/20°C day/night were selected for study. Glasshouse conditions, temperature and air circulation were maintained using the Priva atmospheric regulator and the plants, grown in perforated 25 cm pots were continuously randomized to cancel glasshouse non-uniformity. The seedlings were grouped into four groups. The first group was infested with the FAW and inoculated with the bacterium, Group 2 plants were only infested with the pest, Group 3 plants were only inoculated with the bacterium and finally Group 4 plants were not challenged with the parasites. The plants were grown for 63 days post-parasite exposure and growth and photosynthesis were measured at intervals until the plants were harvested, tubers counted and weighed, and leaf metabolite profiling done on leaves sampled at harvest, 63 days post-parasite exposure.

## Pest infestation and bacterial inoculation

Each of the infested plants was challenged with five first instar larvae of the FAW. The bacterium was grown in Luria-Bertani broth and harvested at the exponential growth stage. The bacterial cells were further suspended in an inoculation buffer (0.0014 M $KH_2PO_4$, 0.0025 M $Na_2HP0_4$, pH 7.00) and the culture was adjusted to $10^8$ CFU. The bacterial inoculum was pressure-filtered through the underside of the leaf. Uninoculated plants were mock-inoculated with the inoculation buffer. Mock-inoculation with the buffer was meant to cancel the effects of the buffer in the bacterial culture. After pest infestation and bacterial inoculation, the plants were individually caged with an insect mesh gauze to contain the pests. The mesh gauze was only removed at days 13, 34 and 47 post experimental treatment to capture the photosynthesis data, observe the pests and capture photographs (not published). Thereafter the caging was removed at termination, 63 days post infestation. The experimental replication was as follows, n=5 for the pest treatment and n=4 for the rest of the treatments.

## Data capture and analysis

At days 13 and 34 post experimental treatment physiological/photosynthesis measurements were captured, and growth was measured at days 20, 27, and 34 post parasite infestation. Finally, the plants were harvested 63 days post treatment, the tubers were weighed, counted and tubers from sample plants are shown on Fig 3. Seven-day intervals were chosen from day 13 post-parasite exposure. We were careful not to disturb the pest on the first 13 days and from day 13 we selected days 20, 27 and 34 as data capture and observation points, and this was consistent with our pre-determined seven-day interval between data capture points. Harvest at day 63 was the point at which tubers had formed, and it was not pre-determined but was decided after observing foliage changes. From previous unpublished work we observed that growth and physiology responses take place during the period we selected for data capture. Physiological parameters measured were photosynthesis rate, stomatal conductance, transpiration efficiency, the ratio of intercellular $CO_2$

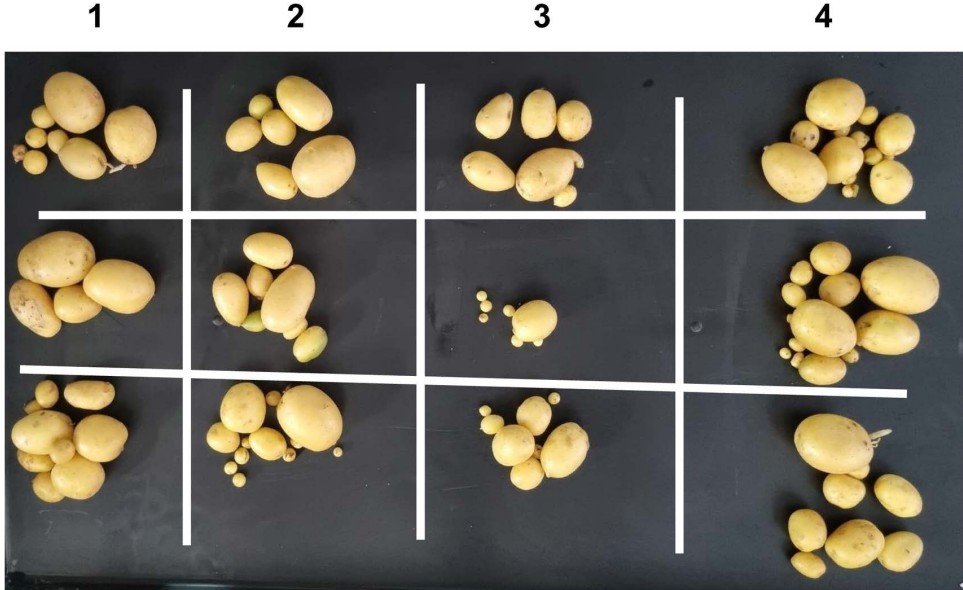

**Fig 3. Potatoes harvested at the 63rd day post-inoculation.** Column 1 (not exposed to parasites), column 2 (inoculated with *Pseudomonas syringae* pv. *tomato*), column 3 (infested with *Spodoptera frugiperda*) and column 4 (infested with *S. frugiperda* and inoculated with *P. syringae* pv. *tomato*). Infestation with either parasite significantly reduced total tuber weight.

concentration to ambient $CO_2$ concentration (Ci/Ca) and water use efficiency. Duplicate measurements of photosynthesis were done using the LI-COR Photosynthesis System (Li-6400, Li-Cor Inc. (Lincoln, NE, USA)) and were taken under full sun. Pest feeding data was not collected and no larvae were observed when the data was captured 13 and 34 days post experimental treatment. All growth, physiology and tuber data were used for the correlation analysis but only the plant height and stem diameter data captured at 34 days post parasite exposure was consider for assessing treatment differences as they appear on Table 1. Data was analyzed using analysis of variance aided with use of the software STATISTICA version 10; significant differences (p = 0.05) were determined using the Duncan multiple range test to separate the means of the measured parameters (Table 1). The measurements were further correlated by Pearson correlation.

## LC-MS metabolite analysis

Liquid chromatography-quadrupole time-of-flight tandem mass spectrometry (LCMS-9030 qTOF, Shimadzu Corporation, Kyoto, Japan) was used to profile metabolites from leaf extracts from the potato plants. The chromatographic separation was conducted at 55°C using a Shim-pack Velox C18 column with dimensions of 100 mm × 2.1 mm and a particle size of 2.7 μm (Shimadzu Corporation, Kyoto, Japan) [14]. A standard injection volume of 5 μL was employed for all samples, which were subsequently subjected to analysis using a binary mobile phase. The mobile phase consisted of solvent A (0.1% formic acid in Milli-Q HPLC grade water) (Merck, Darmstadt, Germany) and solvent B (UHPLC grade methanol with 0.1% formic acid) (Romil Ltd., Cambridge, United Kingdom) [14]. The initial conditions consisted of an isocratic elution with 10% B for three minutes. This was followed by a gradual increase in solvent B to reach 60% over three minutes. Subsequently, the proportion of solvent B was further increased to 90% over three minutes and maintained at this level for one minute. To restore the system to its initial state for the next injection, the conditions were then adjusted to 60% B within one minute and held constant for an additional one minute. This procedure facilitated the re-establishment of equilibrium within the column. The chromatographic analysis was conducted employing a qTOF high-definition mass spectrometer, which was configured to utilize negative electrospray ionization for data acquisition. The mass spectrometer conditions

Table 1. Growth and Photosynthesis Measurements 13 days (underlined) and 34 days After Potato Plants Were Infested With The Fall Armyworm and Inoculated With *Pseudomonas syringae* pv. *tomato*. Letters Accompanying The Values Indicate Significant Differences At ($p<0.05$). Analysis of Variance Was Done Using STATISTICA Software 10; and Significant Differences Were Determined by Duncan Multiple Range Test to Separate the Means of the Parameters Measured.

| Treatment | Photosynthesis rate | Stomatal conductance | Transpiration efficiency | Ratio of intercellular to ambient carbon dioxide concentration | Water use efficiency | Plant height | Stem diameter | Number of tubers | Total tuber weight |
|---|---|---|---|---|---|---|---|---|---|
| | A | Gs | E | Ci/Ca | WUE | | | | |
| | $\mu mol(CO_2)m^{-2}.s^{-1}$ | $Mol(H_2O)m^{-2}.s^{-1}$ | $\mu mol(CO_2) mol^{-1}air^{-1}$ | $\mu mol (CO_2) m^{-2}.s^{-1}$ | $\mu mol (CO_2) m^{-1}.H_2O$ | cm | mm | | |
| Insect + Pathogen (n=5) | 5.88±0.64b <br> 8.07±0.21b | 0.13±0.02b <br> 0.13±0.01a | 3.91±0.53b <br> 3.39±0.26a | 0.64±0.03b <br> 0.66±0.02ab | 55.31±5.52a <br> 68.69±7.62a | 60.60±6.52b | 6.68±0.55ab | 10.60±0.93a | 167.60±22.43a |
| Insect (n=4) | 6.45±0.77b <br> 7.21±0.55ab | 0.12±0.02b <br> 0.16±0.03a | 3.53±0.42b <br> 3.88±0.56a | 0.63±0.02b <br> 0.72±0.03a | 57.70±4.36a <br> 64.81±17.39a | 39.00±5.67a | 5.18±0.59a | 6.75±0.48a | 93.25±20.45b |
| Pathogen (n=4) | 6.41±0.71b <br> 5.96±0.69a | 0.17±0.01b <br> 0.14±0.02a | 4.72±0.31b <br> 3.74±0.35a | 0.74±0.01a <br> 0.71±0.02ab | 37.82±1.92b <br> 44.84±4.19a | 62.75±2.66b | 6.95±0.46b | 7.50±1.94a | 141.00±11.82ab |
| Control (n=4) | 10.46±0.90a <br> 7.59±0.60b | 0.36±0.03a <br> 0.13±0.03a | 7.63±0.19a <br> 3.41±0.47a | 0.78±0.02a <br> 0.64±0.03b | 29.86±2.59b <br> 74.24±15.99a | 65.75±5.11b | 6.64±0.47ab | 7.00±1.08a | 137.75±23.39ab |

encompassed the following parameters: nebulization, interface voltage (3kV), interface temperature (300°C), dry gas flow (0.45 L/min), detector voltage (1.8 kV), heat block (400°C), DL (280°C), and flight tube (42°C) temperatures. The fragmentation of ions was conducted by employing argon gas as the collision medium, following the methodology adopted by Ramabulana et al. [14].

## Multivariate data analysis

The mass spectrometry data obtained from the LCMS-9030 qTOF was pre-processed using the cloud-based bioinformatics program, XCMS Online., which employed HPLC/UHD-qTOF parameters. The centWave feature detection method was employed, with a maximum threshold of 15 ppm and a signal-to-noise ratio of 6. Additionally, prefiltering was conducted based on intensity and noise levels, with thresholds set at 100 and 3, respectively. The approach employed for correcting retention time was obiwarp in conjunction with profStep. The alignment process utilized a minimum proportion of samples of 0.5 and a width of 0.015 m/z. The data was analysed using the Kruskal-Wallis statistical test, which generated a Microsoft Excel table consisting of 8234 distinct attributes. This table was subsequently transposed and saved as a comma-delimited (CSV) file. The CVS table file was thereafter uploaded to the internet-based platform MetaboAnalyst v5.0, where data was normalised by log transformation (base 10) and Pareto scaling was performed before principal component analysis (PCA), partial least squares discriminant analysis (LS-DA), sparse PLS-DA and orthogonal PLS-DA (Figs 4–9). Metabolites were annotated and log2fold change values were calculated (Supplementary S4 Table). Only metabolites with VIP scores greater than one were selected for inclusion on Supplementary S4 Table. The higher the VIP score the more the metabolite contributes to the chemometric projection. Although of significance, the log2fold change was not used for selecting the metabolites to be included. Heatmaps comparing metabolite quantities between the treatment and the control were generated (Supplementary S1, S2, S3 and S4 Figs).

## Results and discussion

Although studies on the effect of a parasite, pest or pathogen, on the plant are commonplace and mostly assert that the plant responds genetically, biochemically and physiologically to the parasite, new twists to these studies bring new

**Fig 4. Computed principal component analysis (PCA) showing separated metabolic features of potato leaves infested with the fall armyworm (FAW) (*Spodoptera frugiperda*), leaves inoculated with the pathogenic bacterium *Pseudomonas syringae* pv. *tomato* (BD2110), leaves double treated with FAW and BD2110, and the control leaves which were non-infested/uninoculated (Control).** Analysis with the LCMS-9030 qTOF was done on leaves sampled 63 days after exposure to the parasites. PCA 1 and PCA2 explain 8.1% and 32.7% of the variation respectively. No definite clustering resulted from the analysis.

dimensions in understanding plant-parasite interactions. Unlike in some previous studies which assessed the response of the plant within 72 hours post exposure of the plant to the parasite [15–17] the current study extended the period between parasite exposure and first assessment because it was anticipated that the physiological response required a longer period and that assessment closer to infestation would disturb pest feeding. Selection of intervals for plant observation and data capture were also consistent with observations from our previous unpublished works that growth and physiology responses are likely from 13 days post parasite exposure. In the current study it was therefore discovered that infestation of potato with the FAW and inoculation with *P. syringae* pv. *tomato* had a profound effect on plant growth as well as on physiological measurements and the metabolome. Plants which were exposed to the FAW alone were shorter than plants which were subjected to the other treatments, namely, the joint FAW-*P. syringae* pv. *tomato* treatment, *P. syringae* pv. *tomato* alone and the non-treated control. Compared to the FAW-only treatment, the height of plants which were treated with both the

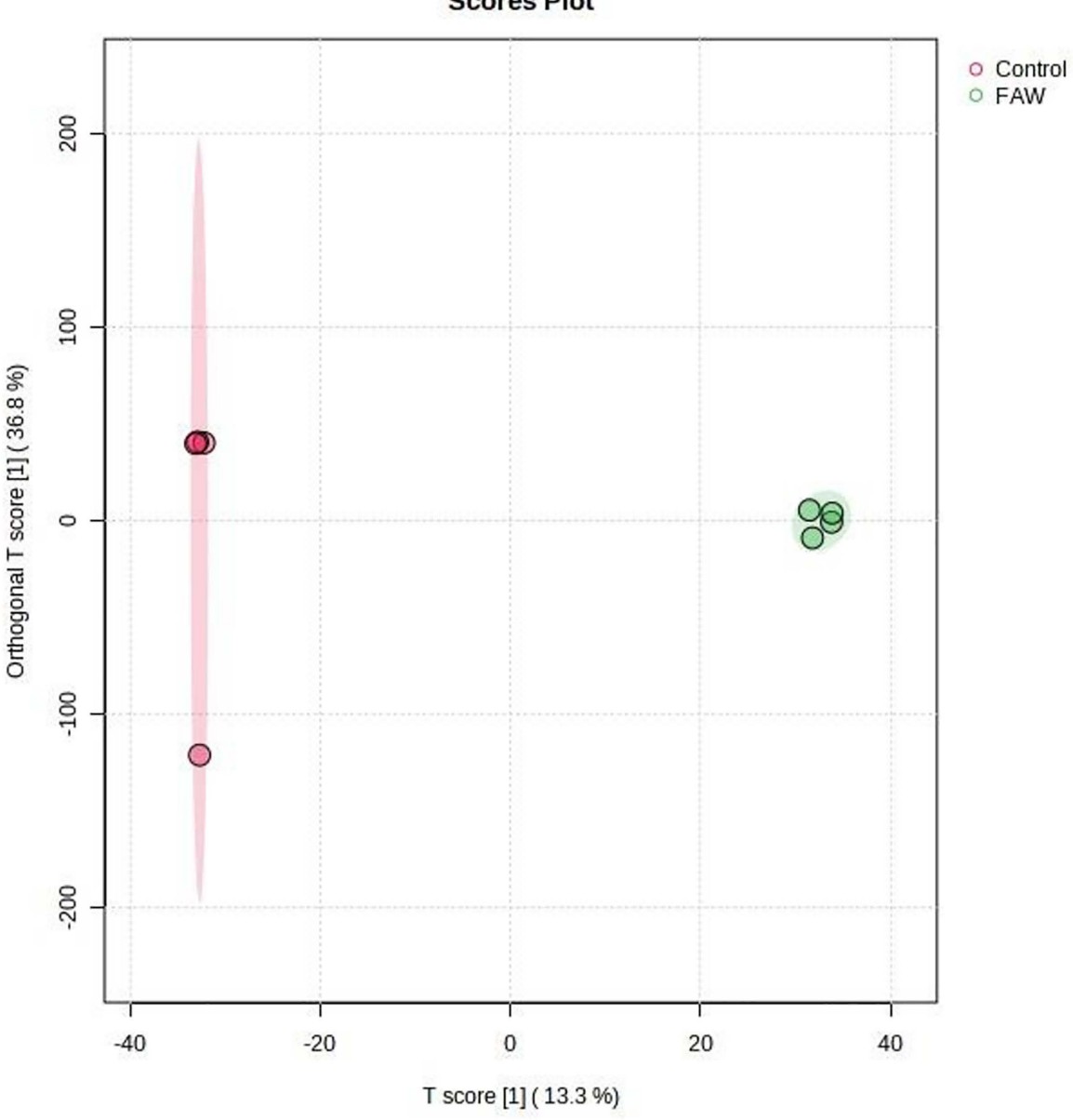

**Fig 5. Computed orthogonal Partial Least Squares – Discriminant Analysis (OPLS-DA) showing separated metabolic features of potato leaves infested with the fall armyworm (FAW) (*Spodoptera frugiperda*), and the control leaves which were non-infested/uninoculated (Control).** Analysis with the LCMS-9030 qTOF was done on leaves sampled 63 days after exposure to the parasites. The illustration shows a clear chromatographic separation between the FAW infested treatments and the non-infested control.

FAW and *P. syringae* pv. *tomato* was not reduced, a hint that the bacterium deters feeding by the arthropod. The FAW-only treatment reduced stem diameter compared to the *P. syringae* pv. *tomato* alone treatment. The number of tubers per plant was unaffected whereas the total tuber weight was reduced by the pest administered alone, and boosted in the joint parasite treatment. This strengthens the perceived notion of antagonism the bacterium had on the pest. All the parasite infestation treatments reduced photosynthesis in the first assessment, 13 days post parasite exposure, but only the bacterium,

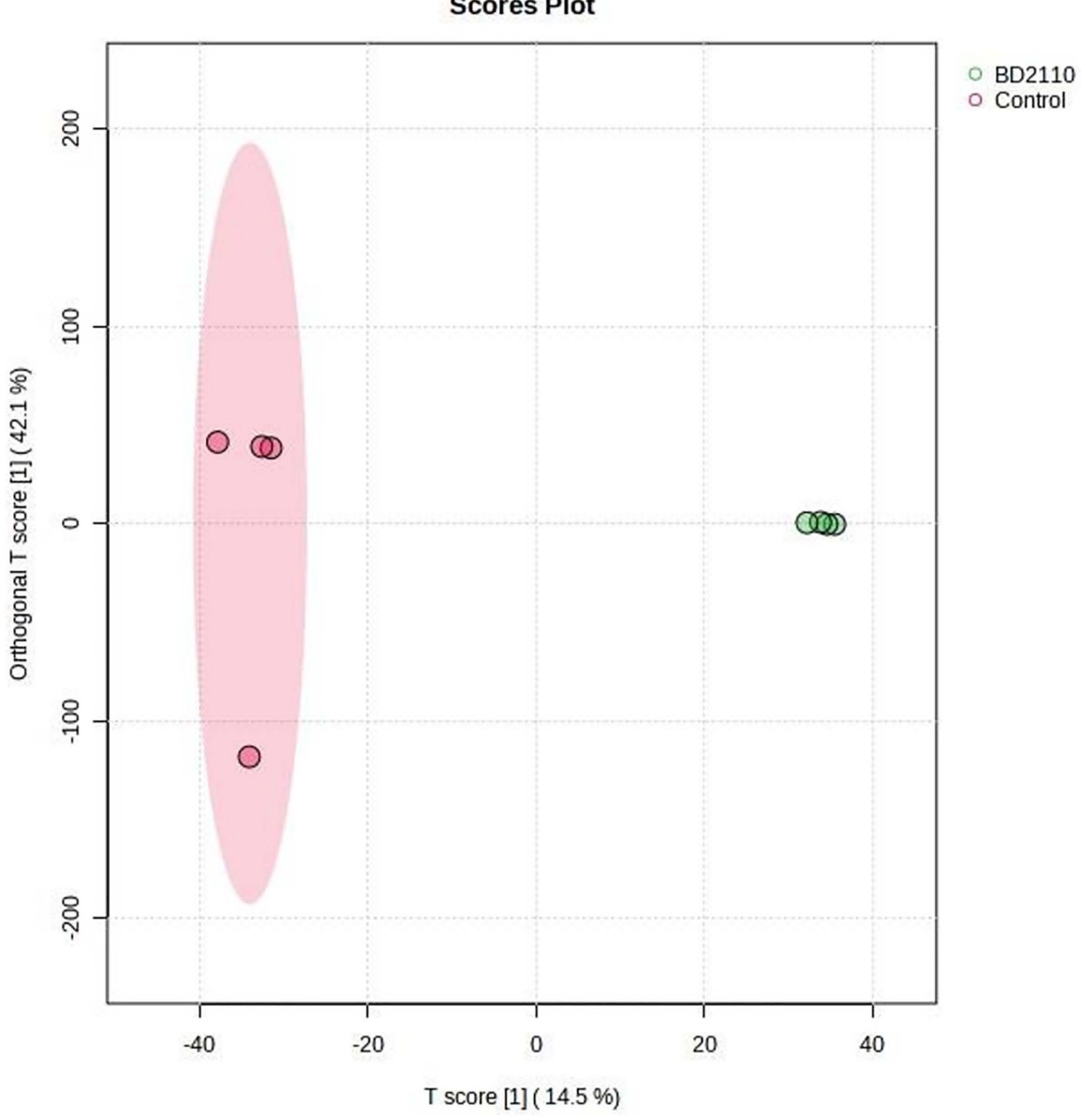

**Fig 6. Computed orthogonal partial least squares – discriminant analysis (OPLS-DA) showing separated metabolic features of potato leaves inoculated with the pathogenic bacterium *Pseudomonas syringae* pv. *tomato* (BD2110), and the control leaves which were uninoculated (Control).** Analysis with the LCMS-9030 qTOF was done on leaves sampled 63 days after exposure to the parasites. The illustration shows a clear chromatographic separation between the BD2110 treatments and the uninculated control.

administered alone, reduced photosynthesis 34 days post treatment. All the treatments reduced stomatal conductance 13 days post treatment but no effect on stomatal conductance was recorded at the 34-day mark. Transpiration efficiency was reduced by all the treatments at 13 days post treatment, and there was no effect at 34 days. Only the two pest treatments reduced Ci/Ca at 13 days post parasite exposure, and Ci/Ca was heightened in the FAW-only treatment 34 days post parasite exposure, and both pest treatments increased water use efficiency 13 post treatment and no treatment affected it at 34 days post treatment. Results on the effect of the treatments on the plant appear on Table 1, Supplementary S1, S2 and S3

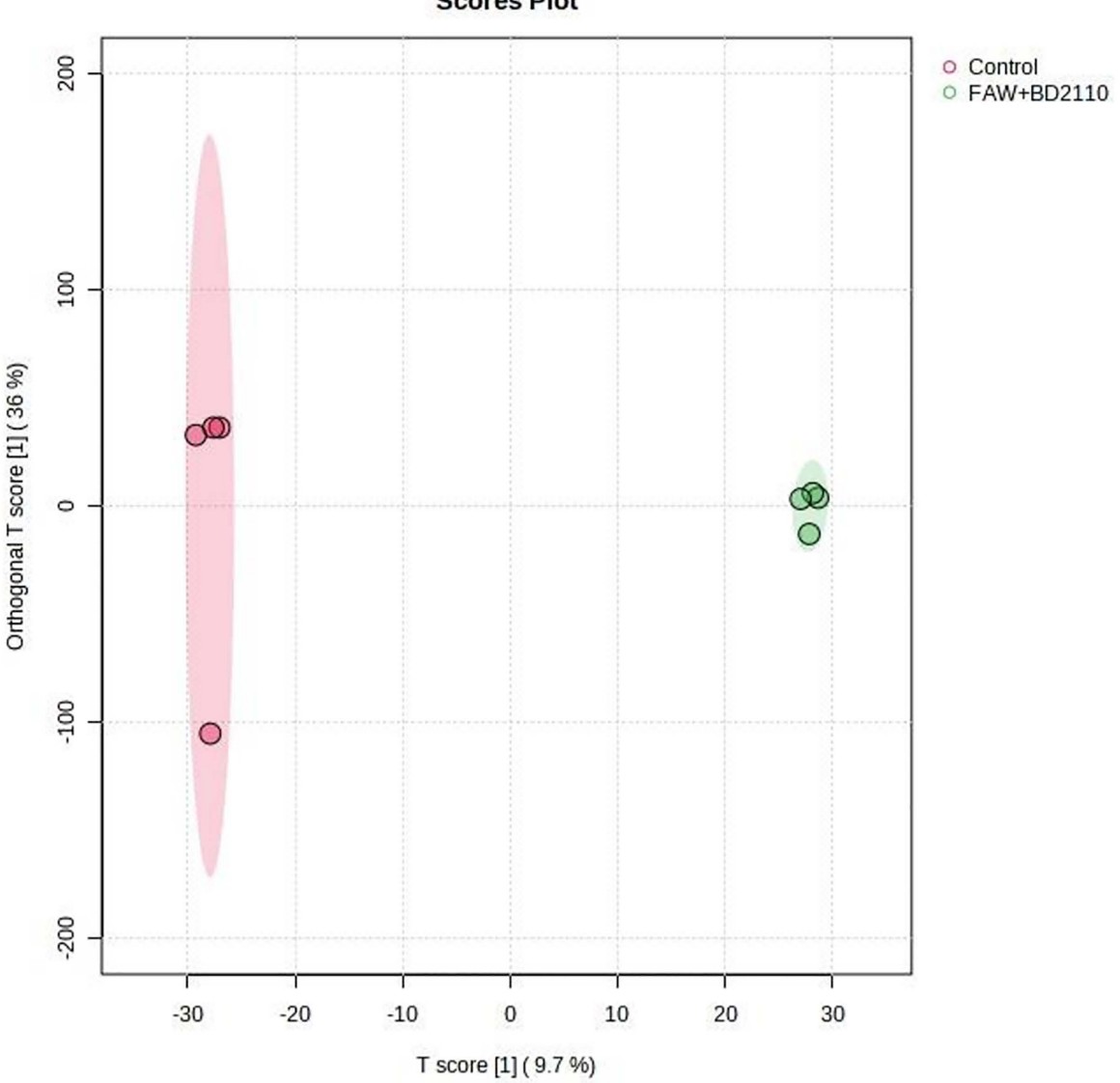

**Fig 7. Computed Orthogonal Partial Least Squares – Discriminant Analysis (OPLS-DA) showing separated metabolic features of potato leaves double treated with fall armyworm (FAW) (*Spodoptera frugiperda*) and *Pseudomonas syringae* pv. *tomato* (BD2110), and the control leaves which were non-infested/uninoculated (Control).** Analysis with the LCMS-9030 qTOF was done on leaves sampled 63 days after exposure to the parasites. The illustration shows a clear chromatographic separation between *P. syringae* pv. *tomato* strain inoculated plus FAW treatments and the uninoculated control.

Tables. Correlation determination of measured parameters revealed interconnections between aspects of growth and physiology, albeit a lack of strong correlations between growth and physiological parameters. Plant height and total tuber weight were strongly correlated (Pearson correlation coefficient = 0.72), plant height was also strongly correlated with stem diameter (0.73). Photosynthesis rate was strongly correlated with transpiration efficiency (0.89) and stomatal conductance (0.81). Stomatal conductance was strongly correlated with Ci/Ca ratio (0.77), transpiration efficiency (0.92) and Ci/Ca ratio (0.77). Transpiration efficiency was negatively correlated with water use efficiency (−0.80) and Ci/Ca (0.76) and finally Ci/Ca was strongly and negatively correlated with water use efficiency (−1.0). These are correlation coefficients calculated from the

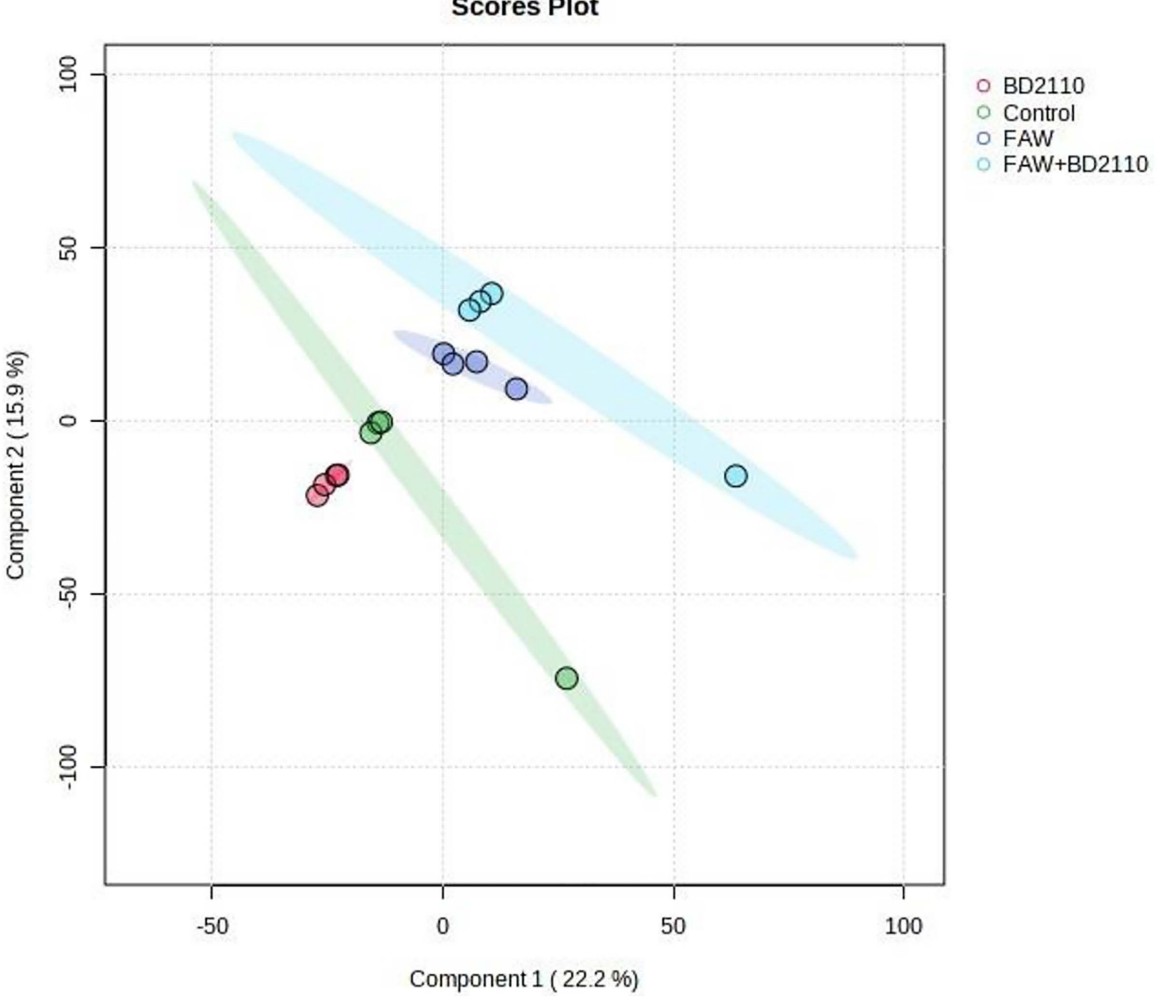

**Fig 8. Computed partial least squares discriminant analysis (PLS-DA) showing separated metabolic features of potato leaves infested with the fall armyworm (FAW) (*Spodoptera frugiperda*), leaves inoculated with the pathogenic bacterium *Pseudomonas syringae* pv. *tomato* (BD2110), leaves double treated with FAW and BD2110, and the control leaves which were non-infested/uninoculated (Control).** Analysis with the LCMS-9030 qTOF was done on leaves sampled 63 days after exposure to the parasites.

first gas exchange readings however second reading correlation coefficients follow a similar trend with transpiration efficiency still correlating strongly with stomatal conductance (0.98), water use efficiency with stomatal conductance (−0.8) and transpiration efficiency (0.87). The Pearson correlation coefficients are presented on Fig 10.

Potato is among the crops grown by both subsistence and commercial farmers due to its nutritional value. Due to its value, there is a constant need for sustainable production of potatoes. Among the challenges faced by this crop is attack by pests and pathogens [18]. The FAW larvae punch holes on leaves, and severe damage is characterized by a skeleton appearance of the leaf, and with severe damage, only the midrib and major veins can remain on the leaf, and this occurs mostly on its preferred host, maize and to a less extent potato [7,19]. The larval population of this pest consumes the foliage of plants to the maximum extent possible when it increases [20]. Studies have shown that this pest can cause significant damage to yield in some crops such as peanuts, barley and wheat by 78, 80%, and 90%, respectively [21,22], including loss of photosynthetic area, impaired reproduction, direct damage to grain, lodging, and structural damage in the

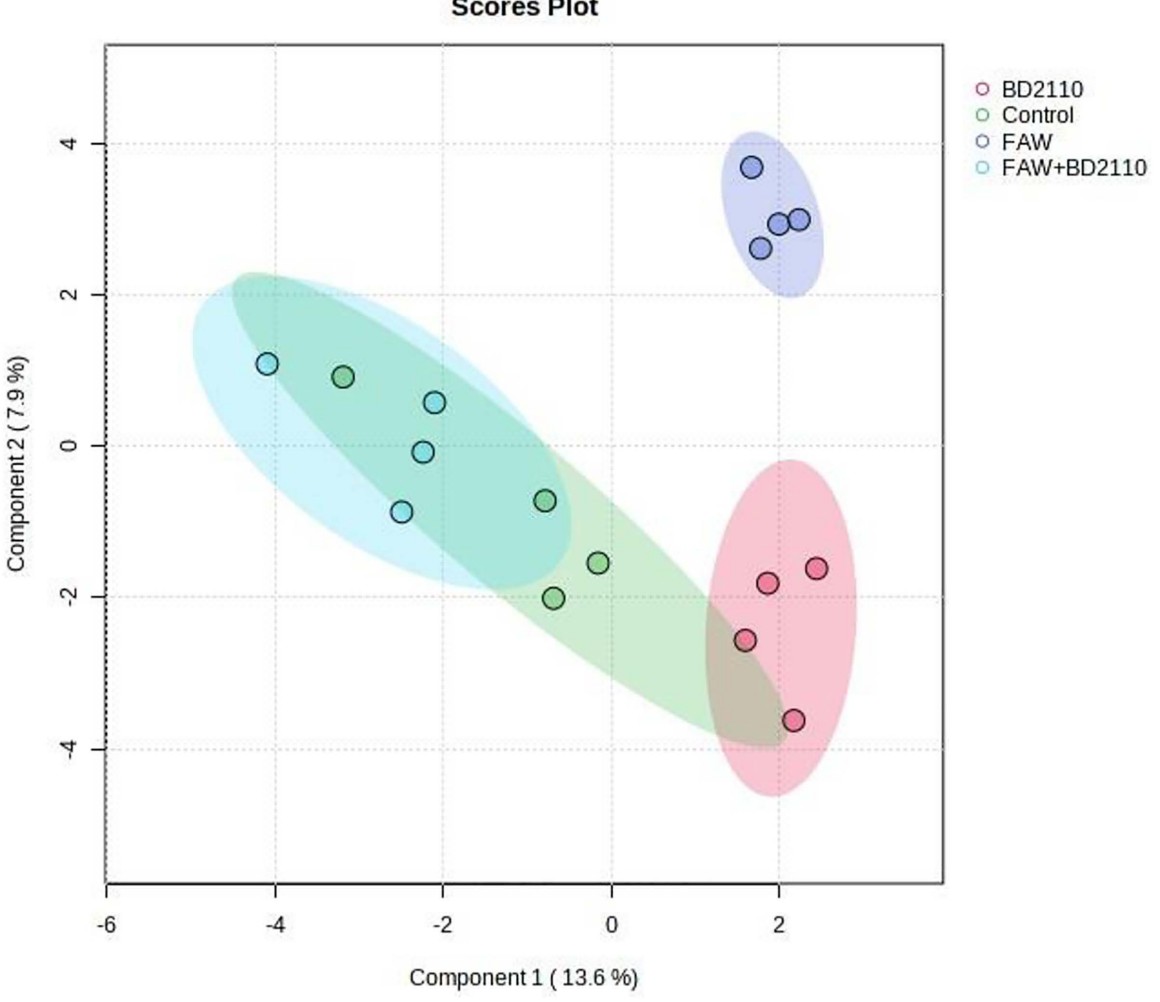

**Fig 9. Computed spatial partial least squares discriminant analysis (SPLS-DA) showing separated metabolic features of potato leaves infested with the fall armyworm (FAW) (*Spodoptera frugiperda*), leaves inoculated with the pathogenic bacterium *Pseudomonas syringae* pv. *tomato* (BD2110), leaves double treated with FAW and BD2110, and the control leaves which were non-infested/uninoculated (Control).** Analysis with the LCMS-9030 qTOF was done on leaves sampled 63 days after exposure to the parasites. The control and the double treatment overlapped.

whorl [23]. In addition, plants that are infested by the FAW react by producing defense metabolites [24]. Unfortunately, this process may lead to a decrease in crop yield [24]. On the other hand, *P. syringae* pv. *tomato* primarily targets the aerial parts of plants, including leaves and fruits, where it can multiply and produce harmful toxins that damage plant tissues. Infected plants typically exhibit distinct symptoms, such as the formation of lesions or discoloration on diseased leaves and necrotic spots on diseased fruits [25]. Additionally, the reduced leaf photosynthetic area and lower fruit quality are common consequences of infection. In this current study, we observed the reduction of growth parameters (plant height and stem diameter) and physiological parameters such as the rate of photosynthesis, stomatal conductance, transpiration efficiency, Ci/Ca, water use efficiency, under FAW and *P. syringae* pv. *tomato* infestation with variation in the level of impact of the two plant parasites. Under bacterial and worm infestation, plants tend to adjust diverse physiological parameters to enable them to survive. Chimweta et al. [23] observed changes in leaves of maize, delay in plant growth, reduction of photosynthesis and a delay in reproduction which finally lead to reduction of plant yield under FAW infestation.

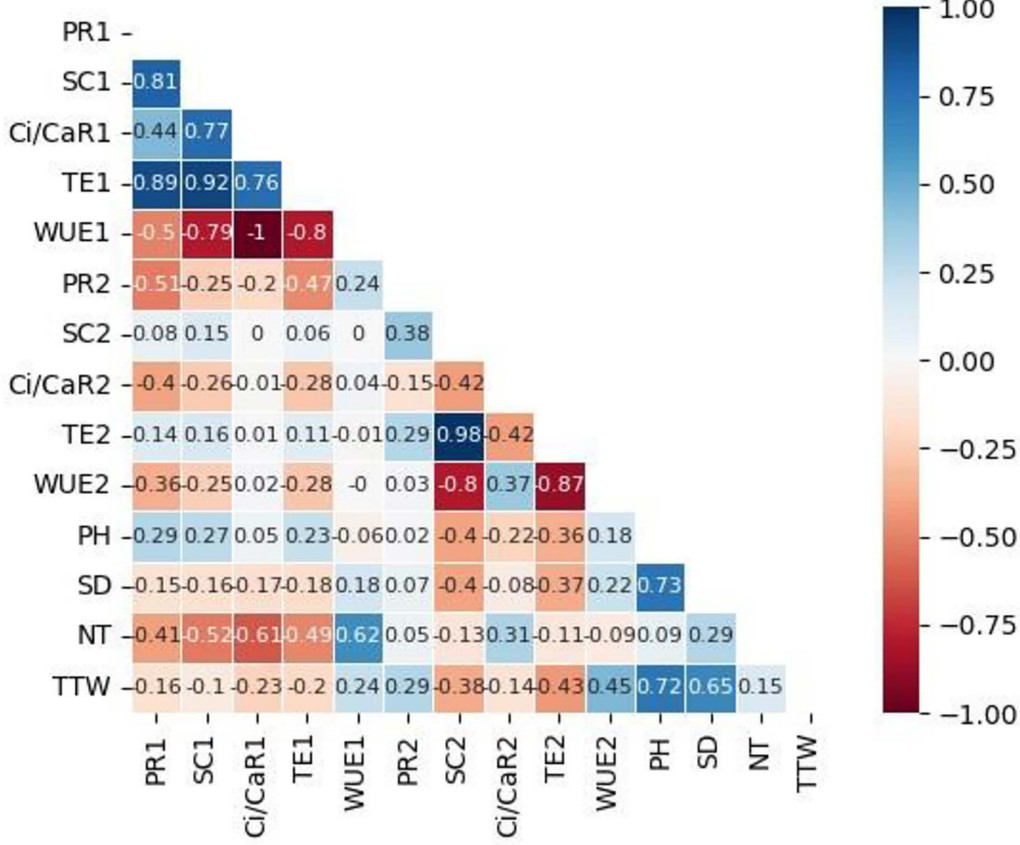

**Fig 10. Correlation heatmap showing Pearson correlation values between growth parameters, photosynthesis measurements taken 13 days and 34 days after infestation of potato leaves with the fall armyworm,** *Spodoptera frugiperda,* **and inoculation with the pathogen** *Pseudomonas syringae* **pv.** *tomato.* The parameters were photosynthesis rate (PR), stomatal conductance (SC), transpiration efficiency (TE), the ratio of intercellular to ambient carbon dioxide concentration (Ci/Ca), water use efficiency (WUE), plant height (PH), stem diameter (SD), number of harvested tubers (NT) and total tuber weight (TTW). The parameters have a number 1 for readings taken at 13 days and a number 2 for readings taken 34 days post parasite exposure.

Similar to the present study, the plant height and yield of potato decrease (compared to the joint parasite treatment) under FAW infestation. The reduction in the yield of potato under FAW infestation can be attributed to feeding by the larvae. Congruent with our findings Zhao et al. [26] and Acharya et al. [4] also reported the devastating impact of the FAW on potatoes especially on growth and physiological function. Regarding the pathogen *P. syringae* pv. *tomato*, its close relative, *P. syringae* pv. *tabaci* reduces the net photosynthetic rate in tobacco leaves [27]. Like our findings, the bacterium showed reduced the rate of photosynthesis as well as other measured parameters of the potato plant. This could be attributed to the different plant and bacterium species used. Stephens and Westoby [28] showed an interrelationship between reduction in plant biomass, leaf area, stem diameter and root growth due to insect feeding. Day et al. [29] reported that infestation by the FAW causes serious damage to the fruit, thereby resulting in premature fruit drop and fruit rot in tomato and pepper. This finding can be attributed to the fact that FAW infestation impairs productivity [30–32], which in turn reduces the amount of food available for household consumption. In the current study tuber weight was reduced only when FAW was administered individually. Combination of the FAW with the bacterium did not diminish tuber weight boosting the emergent view that the bacteria may be inhibiting the pest. Stomata play a pivotal role in how plants respond to biotic stresses. While our understanding of stomata's role in plant stress is primarily centered on abiotic stressors and

plant-pathogen interactions, it is still unclear how stomata influence plant-herbivore interactions [33]. Stomata are intricately linked to various interconnected physiological processes in plants, and any changes in stomatal dynamics caused by any infestation could have significant impacts at the cellular and organismic levels. Stomata are among the most critical structures in land plants, serving as microscopic gates that facilitate the exchange of carbon dioxide ($CO_2$) and water vapor between plants and the atmosphere [34]. As essential regulators of photosynthesis [35] and transpiration, stomata control gas exchange, making them indispensable for plant survival. Although the detailed signaling pathways involved in stomatal sensing and responses to environmental cues have been previously reviewed by Assmann and Jegla [36], the role of stomata in plant-herbivore interactions remains largely unclear. Pathogens have developed various tactics to manipulate stomata for easier access to the plant. For instance, the bacterial pathogen, *Pseudomonas syringae*, produces a toxin called coronatine, which prevents stomatal closure in a COI-dependent manner [37,38]. COI is a subunit of an E3 ubiquitin ligase [39]. Stomatal closure is considered one of the leading causes of herbivore-induced photosynthetic inhibition, which is supported by various studies [40–42]. This phenomenon may also be part of the defense mechanism employed by plants. The closing of stomata in response to herbivory as well as any infestation can regulate various photosynthesis-related processes in plants, such as defense [43]. Although photosynthesis produces essential molecules for the synthesis of defense-related compounds [44,45], the inhibition of growth and photosynthesis often leads to an enhancement in defense [46,47]. Stomatal conductance plays a pivotal role in the regulation of the rate of photosynthesis, transpiration rate, $CO_2$ exchange and other physiological parameters. In the current study, as expected, both the FAW and *P. syringae* pv. *tomato* reduced stomatal conductance as well as transpiration efficiency when compared to the control. This was only the case with 13 days post parasite exposure, and no effect was detected 34 days post treatment, and therefore further work is required to investigate a possible adaptation by the plant after exposure to the parasites. Reduction of the stomatal conductance could be due to the herbivory of the larvae that could signal diverse metabolic responses in the potato plant resulting in the reduction of the stomatal conductance in potato [39]. The same can be speculated about the effect of the bacterium. Diverse metabolic responses could be a mechanism adopted by potatoes to reduce the impact of the larvae to ensure their survival [45]. Studies have shown that there is a positive relationship between plant height and per-plant biomass production [48–51]. This could be associated with an increase in the rate of photosynthesis as the plant height exposes the leaves to the sun for enhanced photosynthesis. In our findings, the correlation of the measured parameters indicated a strong positive correlation between plant height and fruit weight and stem diameter under infestation conditions. This is congruent with previous studies as mentioned above. This implied that under any environmental stress, plants tend to respond by triggering multiple physiological processes. This is also purported by the study done by Radin et al. [52] which found a strong positive correlation between the rate of photosynthesis and stomatal conductance. Other studies have also shown a strong positive correlation between the rate of photosynthesis and stomatal conductance and transpiration efficiency [53–56] which is congruent to our findings. Tuzet [57] stated that stomatal feedback control regulates both transpiration and photosynthesis, and therefore these processes are closely related. This could account for the strong positive correlation between the rate of photosynthesis and rate of transpiration and stomatal conductance in this study. Hui et al. [58] also observed a strong positive correlation between photosynthesis rate and transpiration efficiency throughout the grain yield stage of wheat flag leaves. Yoshie [59] reported negative correlation between intercellular $CO_2$ concentration and water use efficiency of plant with different microhabitat which was similar with our findings where there was a negative correlation between intercellular $CO_2$ concentration and water use efficiency. This could be attributed to the mechanisms that potatoes adopt to enhance their survival under stressed conditions. When the plant is under stress the secondary metabolism is the process which underlies physiological processes. Changes in secondary metabolites profiles were interrogated in the leaves of the potato plant and the LC-MS metabolite analysis using liquid chromatography-quadrupole time-of-flight tandem mass spectrometry (LCMS-9030 qTOF, Shimadzu Corporation, Kyoto, Japan) revealed shifts due to, largely, the action of the pest and the bacterium. Principal component analysis of the generated LC-MS data could not clearly resolve the treatments whereas OPLSDA showed a clear chromatographic

separation between the control treatment and the metabolome of the FAW treated plants (Fig 5), the bacteria treated plants and the control (Fig 6), FAW + bacteria treated plants and the control (Fig 7). PLS also showed chromatographic separations between all the treatments lumped together (Fig 8) and finally SPLS clearly separated the treatments except for FAW + bacterium and the control which seemed to have overlapping influences. The blurred differences between FAW + bacterium and the control were also evident in the tuber weight data and this prompted the authors of this manuscript to view the interaction between the pest and the bacterium as nondetrimental to the plant, and probably a manifestation of parasite-to-parasite antagonism. However, this and other uncovered phenomena require further investigation. Induction of shifts in metabolite profiles caused by biotic stressors is not new. A plethora of studies have uncovered such shifts but not in potato challenged with FAW and *P. syringae* pv. *tomato* and this remains the unique attempt of this study. In a similar study Abu-Nada et al. [60] discovered abundance of 42 metabolites which were conclusively related to infection of the potato plant by *Phytophthora infestans*. Similarly, Zhu et al. [61] discovered 73 such metabolites which meant *P. infestans* induces these changes in potato. In the current study, we listed 50 metabolites which were either downregulated or upregulated in the bacterium treatment, 22 in the FAW treatment and 34 in the FAW + bacterium treatment (Supplementary S4 Table, Supplementary S1, 2, 3 and 4 Figures). Other studies which uncovered upsetting of the metabolome by pest invasion are that of Errard et al., Errard et al., Grover et al., Palmer et al., Li et al. [62–66], which all discovered metabolomic shifts in tomato, sorghum and switchgrass. From these studies, Errand et al. [63] is most striking. This study found that tomato pests, *Tetranychus urticae* Koch and the aphid *Myzus persicae* (Sulzer), induced the production of metabolite signatures on tomato, singly, and in combination with their predator, *Chrysoperla carnea* (Stephens). Regarding the FAW, Grover et al. [64] discovered that a sugarcane aphid infestation reprogrammes sorghum proteome at one and seven days post infestation, and by expectation this is expected to extend metabolomic reprogramming. Similarly, Palmer et al. [65] discovered a sway in metabolites of switchgrass because of feeding by the FAW. Li et al. [66] also discovered the same phenomenon in sugarcane exposed to feeding by the FAW. Our metabolite shifts discovered in plants which had been exposed to the pest for 63 days unlike the one and seven days in the study of Grover et al. [64]. Although studies on the interaction between the plant and *P. syringae* pv. *tomato* are not widespread the few studies concluded provide sufficient clues on whether this pathogen induces metabolic changes in its host plants or not. In the current study, OPLSDA was able to separate the bacterial treated plants from the control which meant there were marked changes induced by the bacterium in the inoculated plants (Fig 6). SPLS showed an overlapping influence of the FAW + bacterium treatment and the control treatment prompting a suspicion that the bacterium deters the pest. Another Pseudomonas, *P. fluorescens* is a known deterrent of plant stressors, e.g. *Streptomyces scabies* and *Phytophthora infestans* [67]. A plethora of other studies of Pseudomonas effectivity against plant parasites exists. López-Gresa et al. [68] detected a rapid accumulation of certain metabolites in tomato plants challenged with *P. syringae* pv. *tomato*. Among the metabolites downregulated in the current study was rutin which was downregulated with a log2FC of −1.54. In the study of López-Gresa et al. [68] rutin was detected after *P. syringae* infection prompting a role in defense. Rutin is also linked with the interaction between wheat and *Puccinia striiformis* [69,70]. However, the role of rutin in tomato and potato plants infected with *P. syringae* pv. *tomato* may require a deeper investigation. Jelenska et al. [71] discovered a hijack of the chaperone machinery of plants by *P. syringae* clearly indicating a sway in metabolite profiles due to infection with the pathogen. Metabolomics deciphers complex interactions of certain defence-related metabolic changes and the complex can be arduous to comprehend. From the current study we listed 50 metabolites which were either downregulated or upregulated in the bacterium treatment, 22 in the FAW treatment and 34 in the FAW + bacterium treatment (Supplementary S4 Table, Supplementary S1, 2, 3 and 4 Figures). Only metabolites with a VIP score greater than one (1<) were included in Supplementary S4 Table. The greater the VIP score the greater is the chemometric projection contribution by the metabolite. From the long list of metabolites (on Supplementary S4 Table) our discussion is limited to a few metabolites. The first is α-Solanine, α-Solanine is a glycoalkaloid found in species of the nightshade family including potato [72,73]. It has antimicrobial properties such as fungitoxicity [74,75]. Its role in the potato leaves as a downregulated compound in the

current study remains unknown but it unlikely to be boosting resistance as hinted by Gregory, Sinden et al. Sarquis et al. [76–78]. The other metabolite is 6-Gingerol. 6-Gingerol reduces *Pseudomonas aeruginosa* biofilm formation and virulence via quorum sensing inhibition [79,80]. Another compound is laetisaric acid which has demonstrated fungicidal activity [81] and citric acid was proven to suppress the pathogen *Fusarium oxysporum* f. sp. *lycopersici* [82], to reduce the diseases Goos's wilt, leaf blight of corn [83] and limit decay of peach postharvest [84]. The authors of this manuscript concluded that *P. syringae* pv. *tomato* and FAW upset the growth and physiology of the potato plant and both induce metabolic changes in leaves of infected plants. We noted that growth and tuber production were affected and there is variation in the administered treatments. The pest was more severe when administered individually and this pattern appeared in the height of the plant, stem diameter and total tuber weight. The plant height, stem diameter and tuber weight which were unaffected in the plants which were co-challenged with both parasites signalled a possible discord between the bacterium and the pest. This discord led to a cancellation of the parasitic effects of the pest. However, this matter is inconclusive in the current study and therefore requires further investigation. Elucidation of the roles of individual metabolites in response to parasitism by the pest and the bacterium requires another detailed investigation. Several aspects of this study require explanation. Firstly, the potato cultivar Georgina was chosen because of its popularity as a cultivar and no resistance status to the FAW and *P. syringae* pv. *tomato* is recorded. Secondly, potato is not the primary host of the FAW and *P. syringae* pv. *tomato*. However, the combination of the three, the potato and the two parasites had to be unique and it therefore served its purpose of uncovering a new plant-parasite interaction with characteristics not recorded previously. A study by Guo et al. [85] showed that the FAW is able to survive on potato although the preferred host remains maize. In the current study the FAW affected most of the indices measured, however, its impact lessened when it was co-administered with the bacterium. Repeated applications of the larvae would have revealed different results. In the current study the larvae were applied once and their survival checked at day 13 post infestation and in subsequent plant examinations. Only signs of larval feeding were evident, the larvae themselves were not present 13 days post infestation, an indication of pupation. The feeding on the leaves could be regarded as inconsistent nibbling without major damage. This was also evident at day 47 post infestation. At forty seven days post infestation the leaves were still green with minimal damage. Because of the erratic nature of its feeding on the leaves, the authors of this manuscript decided that rather than scoring pest damage, the effect of the pest was best measurable by assessing effect on the growth, physiological indices and the metabolome. Of the bacterial pathogens which have been used to study plant-pathogen interactions, *P. syringae* pv. *tomato* is one of the notably models. Although potato is not the primary host for *P. syringae* pv. *tomato*, interaction with *P. syringae* pv. *tomato* would allow comparison with tomato studies. Moreover, with pressurised inoculation physical barriers are overcome and therefore it matters less if the bacterial pathogen is a known pathogen in the plant-pathogen relationship being studied. Furthermore, metabolomic profiling in this study was undertaken 63 days post parasite application. This is unlike in a plethora of studies, for example [15–17] when plant responses were evaluated within 72 hours of administering the parasites. In the current study the interest was in permanent changes in the metabolomic profile. Studies which investigate direct metabolomic responses to infection assess metabolomic changes within the window of within 72 hours post parasite exposure, or a little longer. In such cases the metabolites of interest are those which respond to overcome infestation and infection. Sampling within 72 hours would disturb pest feeding and the larvae would probably escape from the caging. Since our interest was in permanent metabolite changes we resorted to sample for metabolomic analysis at termination, 63 days after parasite exposure. With such a late sampling point at 63 days post treatment, we were unable to attribute the metabolomic reprogramming to direct defense against the parasites, and implicate such pathways as the known defense pathway, the phenylpropanoid pathway. Metabolomic assessment at 63 days post parasite exposure may also explain why a known defense metabolite, rutin, was not upregulated in our assessment. However, metabolomic analysis at this time point of 63 days proved valuable because the experimental treatments could be differentiated. Ideally, multiple points of sampling would have provided a clearer picture of plant-parasite relations. The highlights of this study are that the parasites induce morpho-physiological changes, and reprogramme the metabolome of

infested and inoculated potato plants. Rather than having additive parasitic effects when combined and administered to the plants, the parasites possibly exert parasite-to-parasite antagonism which requires an additional investigation. Despite the inconclusive aspects of this study, the results obtained in the current study are valuable and add crucial knowledge to our understanding of potato-parasite dynamics. Should the deterrent of the pest by the bacterium be confirmed, a potential biological control product is possible, and this may benefit potato farmers if the FAW becomes devastating. Moreover, given that some parameters correlated strongly with others it is possible to manipulate one parameter to affect another parameter. For an example, total tuber weight depends strongly on plant height and stem diameter. This means a grower may aim for taller and thicker plants to boost yield. Additionally, the number of tubers per plant also depends on water use efficiency which means any agent which boosts water use efficiency will likely affect this important yield parameter, the number of tubers per plant. As various efforts of plant manipulation continue, optimal and best practice combinations will be discovered and this will improve potato productivity. This study helped to confirm the interlinks between plant processes for better understanding of plant processes, especially plants exposed to parasites.

## Supporting information

**S1 Fig. An experiment of infestation/inoculation of potato leaves with *Pseudomonas syringae* pv*. tomato* (BD2110) and the fall armyworm (FAW),* Spodoptera frugiperda*.** An OPLS-DA S-plot utilizing Pareto scaling with mean centering to compare control and inoculated potato leaves. The UHPLC-qTOF-MS (Negative mode) data sets of potato leaf samples compared to control, BD2110, FAW, and BD2110+FAW samples form the basis of the models. A (Control vs BD2110) B (Control vs FAW) C (Control vs FAW+BD2110).
(DOCX)

**S2 Fig. Interactive heatmap analysis profiles of annotated metabolites from potato leaves inoculated with *Pseudomonas syringae* pv*. tomato* (BD2110).** The treatments were *P. syringae* pv. *tomato*, strain BD2110 against the untreated control.
(DOCX)

**S3 Fig. Interactive heatmap analysis profiles of annotated metabolites from potato leaves infested with the fall armyworm (FAW), *Spodoptera frugiperda*.** The treatments were against the untreated control.
(DOCX)

**S4 Fig. Interactive heatmap analysis profiles of annotated metabolites from potato leaves infested/inoculated with *Pseudomonas syringae* pv*. tomato* (BD2110) and the fall armyworm (FAW),* Spodoptera frugiperda*.** The treatments were *P. syringae* pv. *tomato*, strain BD2110+FAW against the untreated control.
(DOCX)

**S1 Table. Growth data (plant height, stem diameter, number of tubers and total tuber weight) of plants infested with *Spodoptera frugiperda* (the fall armyworm, FAW) and inoculated with the bacterium *Pseudomonas syringae* pv*. tomato* (strain BD2110).** The day of the infestation/inoculation was marked day 0 and plants were tracked throughout the 63 days of the experiment. Growth was measured at days 20, 27, and 34 and finally, the plants were harvested at day 63. The initial data was for tracking the plants and only data on days 34 and 63 was used to differentiate the treatments.
(DOCX)

**S2 Table. Physiological data (photosynthesis rate, stomatal conductance, transpiration efficiency, the ratio of intercellular $CO_2$ concentration to ambient $CO_2$ concentration (Ci/Ca ratio) and water use efficiency) of plants infested with *Spodoptera frugiperda* (the fall armyworm, FAW) and inoculated with the bacterium *Pseudomonas*

*syringae* pv. *tomato* **(strain BD2110).** The physiological data was captured 13 days after infestation/inoculation. Two measurements were taken per plant.
(DOCX)

**S3 Table. Physiological data (photosynthesis rate, stomatal conductance, transpiration efficiency, the ratio of intercellular $CO_2$ concentration to ambient $CO_2$ concentration (Ci/Ca ratio) and water use efficiency) of plants infested with *Spodoptera frugiperda* (the fall armyworm, FAW) and inoculated with the bacterium *Pseudomonas syringae* pv. *tomato* (strain BD2110).** The physiological data was captured 34 days after infestation/inoculation. Two measurements were taken per plant.
(DOCX)

**S4 Table. The annotated metabolites in *Solanum tuberosum* L. leaves infested/inoculated with *Pseudomonas syringae* pv. *tomato* (BD2110), *Spodoptera frugiperda* (FAW) and FAW + BD2110 exhibit concentration differences that are statistically significant between untreated (Control) and treated plants (BD2110, FAW and FAW + BD2110).**
(DOCX)

## Acknowledgments

The authors extend their gratitude to the University of South Africa and the University of Venda, the two institutions where the work was performed. Ms. Valery Moloto and Dr. Teresa Goszczynska of the Agricultural Research Council – Plant Health and Protection provided *Pseudomonas syringae* pv. *tomato*.

## Author contributions

**Conceptualization:** Khayalethu Ntushelo.

**Data curation:** Sandra Maluleke, Udoka Vitus Ogugua, Ntakadzeni Edwin Madala, Khayalethu Ntushelo.

**Formal analysis:** Ntakadzeni Edwin Madala, Mbalenhle Precious Mazibuko, Khayalethu Ntushelo.

**Funding acquisition:** Khayalethu Ntushelo.

**Investigation:** Sandra Maluleke, Udoka Vitus Ogugua, Ntakadzeni Edwin Madala, Phumzile Sibisi, Mbalenhle Precious Mazibuko, Robert Sicelo Nofemela, Ebere Lovelyn Udeh, Khayalethu Ntushelo.

**Methodology:** Sandra Maluleke, Udoka Vitus Ogugua, Ntakadzeni Edwin Madala, Phumzile Sibisi, Mbalenhle Precious Mazibuko, Robert Sicelo Nofemela, Khayalethu Ntushelo.

**Project administration:** Khayalethu Ntushelo.

**Resources:** Ntakadzeni Edwin Madala, Robert Sicelo Nofemela, Khayalethu Ntushelo.

**Software:** Udoka Vitus Ogugua, Ntakadzeni Edwin Madala.

**Supervision:** Khayalethu Ntushelo.

**Validation:** Phumzile Sibisi.

**Visualization:** Robert Sicelo Nofemela.

**Writing – original draft:** Ebere Lovelyn Udeh, Khayalethu Ntushelo.

**Writing – review & editing:** Ebere Lovelyn Udeh, Khayalethu Ntushelo.

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
