## [Decision Letter · Decision Letter 0]

19 Jun 2025

Dear Dr. Ntushelo,

Thank you for submitting your manuscript to PLOS ONE. After careful consideration, we feel that it has merit but does not fully meet PLOS ONE’s publication criteria as it currently stands. Therefore, we invite you to submit a revised version of the manuscript that addresses the points raised during the review process.

We look forward to receiving your revised manuscript.

Kind regards,

Arka Pratim Chakraborty, Ph. D.

Academic Editor

PLOS ONE

Journal Requirements: 

 [This study was partially funded by the National Research Foundation of South Africa under the grant TTK170413227119.]. 

Additional Editor Comments:

Dear author

You are advised to rectify the paper as per the reviewer comments. The article can be considered for acceptance after major revision.

With regards

Dr. Arka Pratim Chakraborty

Academic Editor

Assistant Professor

Department of Botany

Raiganj University

Email- arka.botanyrgu@gmail.com

Reviewers' comments:

Reviewer's Responses to Questions

**Comments to the Author**

1. Is the manuscript technically sound, and do the data support the conclusions?

Reviewer #1: Yes

Reviewer #2: Yes

2. Has the statistical analysis been performed appropriately and rigorously?

Reviewer #1: Yes

Reviewer #2: Yes

3. Have the authors made all data underlying the findings in their manuscript fully available?

Reviewer #1: Yes

Reviewer #2: Yes

4. Is the manuscript presented in an intelligible fashion and written in standard English?

Reviewer #1: Yes

Reviewer #2: Yes

Reviewer #1: The manuscript titled “Pseudomonas syringae pv. tomato and the fall armyworm modulate the morphophysiology and the metabolome of potato plants” presents a detailed study investigating the individual and combined effects of Pseudomonas syringae pv. tomato (Pst) and the fall armyworm (Spodoptera frugiperda, FAW) on the morphophysiology and metabolome of potato (Solanum tuberosum) plants. The experimental setup with four treatment groups (control, FAW, Pst, FAW+Pst) and a clear timeline (Fig. 2) is well-structured. The study is well-designed, addressing a gap in understanding how these two biotic stressors interact and affect a major crop. The inclusion of growth, physiological, and metabolomic analyses, along with correlation studies, provides a comprehensive view of plant responses. However, there are several areas where clarity, methodological rigor, and interpretation can be improved to strengthen the manuscript. The manuscript states that uninoculated plants were mock-inoculated with buffer, but it does not clarify whether the buffer infiltration itself caused any physiological or metabolomic effects. This could confound results, especially for Pst treatments. The use of five first-instar FAW larvae per plant is noted, but there is no information on how larval feeding was controlled or monitored (e.g., whether larvae were replaced if they died or stopped feeding, or how feeding damage was quantified). The metabolomic analysis identifies 50, 22, and 34 metabolites affected in Pst, FAW, and combined treatments, respectively, but the manuscript lacks details on how these metabolites were selected or validated. For example, were they identified based on statistical significance, fold change thresholds, or biological relevance?

- Abstract and Introduction:

- The abstract is dense and includes technical details (e.g., specific correlations) that may overwhelm readers. It should focus on key findings and their implications. Emphasizing the interaction between Pst and FAW, key physiological and metabolomic findings, and their significance. Avoid listing all correlations.

- The introduction provides good context but could better justify why Pst was chosen for potato, given its primary association with tomato and Arabidopsis. Strengthen the rationale for studying Pst on potato by citing evidence of its cross-host infectivity (e.g., Arabidopsis, tomato) and clarifying the knowledge gap.

- Materials and Methods:

- Specify the potato cultivar’s susceptibility to FAW and Pst to justify its selection.

- Clarify the timing of metabolomic sampling (e.g., were leaves sampled at day 63, as implied by Figs 4-9?).

- Provide more details on statistical analyses (e.g., assumptions for ANOVA, handling of non-normal data).

- Results and Discussion:

- The results section is overly descriptive, repeating data from tables and figures without synthesizing key trends. For example, the discussion of physiological parameters could be more concise, focusing on significant differences and their biological implications. Avoid repeating table/figure data verbatim; instead, highlight significant trends and refer to visuals. Resolve the discrepancy between Figure 3 and text regarding tuber weight significance.

- The discussion is comprehensive but includes speculative statements (e.g., “the bacterium deters the pest”) without sufficient mechanistic evidence. The overlap in metabolomic profiles between FAW+Pst and control treatments (Fig. 9) is noted but not fully explored.

- Figures and Tables:

- Tables 1 and 2 are informative but could be combined into a single table with subheadings for 13-day and 34-day measurements to save space and improve readability.

- Figure 3 states “no statistically significant differences” in tuber weight, which contradicts the text (Page 24, Line 291) claiming significant reductions by individual parasites. This discrepancy needs resolution.

Interpretation and Conclusions

- Overstatement of Pst deterrence:

- The claim that Pst deters FAW is based primarily on the lack of plant height reduction in co-treated plants and the overlap in metabolomic profiles. However, alternative explanations (e.g., compensatory plant responses, reduced FAW feeding due to plant stress) are not considered.

- Metabolomic Insights:

- The discussion of metabolites like α-solanine, 6-gingerol, and rutin is interesting but lacks depth. For example, the downregulation of α-solanine is noted, but its potential role in plant defense or FAW deterrence is not explored. Expand the discussion of key metabolites, linking their changes to known defense pathways or pest/pathogen interactions. For instance, discuss whether rutin downregulation aligns with Pst’s manipulation of plant defenses (as hinted in López-Gresa et al., 2011).

- Broader Implications:

- The manuscript concludes that the findings add valuable knowledge, but it does not discuss practical implications for potato farming or pest management (e.g., could Pst or related bacteria be used as biocontrol agents against FAW?).

Minor Issues

- Typographical and grammatical errors:

- There are minor errors, e.g., “ineral ratio” (Page 1, Abstract) should be “Ci/Ca ratio,” and “uninoculated” is inconsistently hyphenated (e.g., Pages 20-22). Conduct a thorough proofread to correct typos and ensure consistency in terminology.

Reviewer #2: Dear Authors,

Suggestions for correction are in the attached file.

Too much information about FAW:

The section on the nocturnal behavior of the two FAW biotypes is unnecessary.

Suggestion: Reduce to a concise paragraph.

Best regards,

**Do you want your identity to be public for this peer review?** For information about this choice, including consent withdrawal, please see our Privacy Policy

Reviewer #1: **Yes: ** Ali Mokhtassi-Bidgoli

Reviewer #2: **Yes: ** Marconi Batista Teixeira

---

## [Author Response · Author response to Decision Letter 1]

3 Aug 2025

The review process has helped to strengthen the manuscript. The reviewers were meticulous in doing the review.

---

## [Decision Letter · Decision Letter 1]

12 Oct 2025

Pseudomonas syringae pv. tomato and the fall armyworm modulate the morpho-physiology and the metabolome of potato plants

PONE-D-25-21507R1

Dear Dr. Ntushelo,

We’re pleased to inform you that your manuscript has been judged scientifically suitable for publication and will be formally accepted for publication once it meets all outstanding technical requirements.

Kind regards,

Mayank Anand Gururani

Academic Editor

PLOS ONE

Additional Editor Comments (optional):

Reviewers' comments:

Reviewer's Responses to Questions

**Comments to the Author**

Reviewer #1: All comments have been addressed

2. Is the manuscript technically sound, and do the data support the conclusions?

Reviewer #1: Yes

3. Has the statistical analysis been performed appropriately and rigorously?

Reviewer #1: Yes

4. Have the authors made all data underlying the findings in their manuscript fully available?

Reviewer #1: Yes

5. Is the manuscript presented in an intelligible fashion and written in standard English?

Reviewer #1: Yes

Reviewer #1: The authors have implemented all required revisions, addressed every reviewer comment, and thoroughly proofread the manuscript; it is now finalized and ready for publication.

**Do you want your identity to be public for this peer review?** For information about this choice, including consent withdrawal, please see our Privacy Policy

Reviewer #1: **Yes: ** Ali Mokhtassi-Bidgoli

---

## [Editor Report · Acceptance letter]

PONE-D-25-21507R1

PLOS One

Dear Dr. Ntushelo,

I'm pleased to inform you that your manuscript has been deemed suitable for publication in PLOS One. Congratulations! Your manuscript is now being handed over to our production team.

Kind regards,

on behalf of

Dr. Mayank Anand Gururani

Academic Editor

PLOS One